# An Alternative Application of Magnetic-Activated Cell Sorting: CD45 and CD235a Based Purification of Semen and Testicular Tissue Samples

**DOI:** 10.3390/ijms25073627

**Published:** 2024-03-24

**Authors:** Péter Czétány, András Balló, László Márk, Attila Török, Árpád Szántó, Gábor Máté

**Affiliations:** 1Urology Clinic, University of Pécs Clinical Centre, 7621 Pécs, Hungary; czetany.peter@pte.hu (P.C.); ballo.andras@pte.hu (A.B.); mate.gabor@pri.hu (G.M.); 2National Laboratory on Human Reproduction, University of Pécs, 7624 Pécs, Hungary; laszlo.mark@aok.pte.hu; 3Pannon Reproduction Institute, 8300 Tapolca, Hungary; drtoroka@t-online.hu; 4Department of Analytical Biochemistry, Institute of Biochemistry and Medical Chemistry, University of Pécs Medical School, 7624 Pécs, Hungary; 5MTA-PTE Human Reproduction Scientific Research Group, 7624 Pécs, Hungary

**Keywords:** annexin V, assisted reproduction, erythrocyte, leukocyte, magnetic activated cell sorting, magnetic bead, sperm

## Abstract

Magnetic activated cell sorting (MACS) is a well-known sperm selection technique, which is able to remove apoptotic spermatozoa from semen samples using the classic annexinV based method. Leukocytes and erythrocytes in semen samples or in testicular tissue processed for in vitro fertilization (IVF) could exert detrimental effects on sperm. In the current study, we rethought the aforementioned technique and used magnetic microbeads conjugated with anti-CD45/CD235a antibodies to eliminate contaminating leukocytes and erythrocytes from leukocytospermic semen samples and testicular tissue samples gained via testicular sperm extraction (TESE). With this technique, a 15.7- and a 30.8-fold reduction could be achieved in the ratio of leukocytes in semen and in the number of erythrocytes in TESE samples, respectively. Our results show that MACS is a method worth to reconsider, with more potential alternative applications. Investigations to find molecules labeling high-quality sperm population and the development of positive selection procedures based on these might be a direction of future research.

## 1. Introduction

A global crisis of human fertility is developing since the mid-20th century, currently 1 in every 6 people is affected according to the World Health Organisation (WHO) [1]. For this population, assisted reproduction techniques (ARTs) offer a chance to have their own biological offspring since its first success in 1978 [2]. Despite their worldwide use, the efficacy of ARTs is still relatively low: 37.3% of ART cycles resulted in live-birth deliveries in the USA in 2021 [3].

The male factor is present in 30 to 50% of infertile couples [4]. With ARTs (especially with intracytoplasmic sperm injection (ICSI)) several natural mechanisms (vaginal acidic pH, cervical mucus, cellular immune response, sperm-epithelial interactions in the caudal isthmus of the Fallopian tube, sperm-zona pellucida interactions [5,6]), resulting in selection of the sperm subpopulation with highest fertilization potential, are bypassed. Thus, presumably with the isolation and use of these particular cells, it might raise the success rate of assisted reproduction. For this purpose, many sperm selection techniques have been developed: the swim-up test (SU) and the densitygradient centrifugation (DGC) are routinely used. SU is one of the simplest, fastest, cheapest and most commonly used methods. It is based on the migration of the healthy spermatozoa. The sample is placed in a tube placed in a 45° angle with a layer of culture medium on top. After certain time of incubation (usually 1 h) a motile fraction of sperm can be harvested from the upper layer, with high probability of normal morphology and higher DNA integrity. The drawback of the technique is the low percentage (circa 5–10%) of cells retrieved. DGC is based on different densities of normal/abnormal sperm, contaminating cells and debris in semen samples. A density gradient is formed from the colloidal suspension of silica particles, and after centrifugation the lower phase contains the better-quality sperm subpopulation, while the abnormal spermatozoa, other cells, non-cellular contaminants can be found in the upper phase. One possible disadvantage can be the formation of reactive oxygen species (ROS), although its chance could be minimized with the application of the lowest possible centrifugal force (generally 300× *g* for 20 min). In some specific, more complicated case (severe oligozoospermia, high viscosity) these conventional selection techniques could be inefficient [7]. In these occasions, the advanced sperm selection methods (e.g., microfluidic devices, physiological intracytoplasmic sperm injection (PICSI)) can improve the results, though these are sporadically applied depending on their accessibility and preferences of the fertility centre. None of these methods has been proven to be unequivocally beneficial in large studies, so their routine application cannot be recommended yet [8,9]. In this regard further studies are needed. In the clinical practice, sperm selection is still based on the subjective (mainly morphological) evaluation and expertise of the embryologist.

The methodology of magnetic activated cell sorting (MACS) was published first in 1995 by Pesce and De Felici [10], to separate primordial germ cells of mouse embryos from somatic cells. It is commonly used in immunology, oncological research, neuroscience and stem cell research [11,12,13,14,15,16]. Dunlap et al. [17] used the method to identify immunoglobulin (Ig) G-bound bacteria from bronchoalveolar lavage samples to study lung microbiome. The technique was applied to identify IgG-bound bacteria participating in the pathogenesis of inflammatory bowel disease (IBD) [18]. Different cell selection strategies exist, the predominant method is based on cell surface markers using ligands targeting them. These ligands are most frequently antibodies, but synthetic molecules like aptamers and peptids have been developed as well. Label-free magnetic cell sorting is also a viable option in case of specific cells containing paramagnetic materials (e.g., hem molecule in erythrocytes) providing intrinsic susceptibility for magnetophoresis. Shamloo and his colleagues developed a two-step microfluidic device which could separate circulating tumor cells from erythrocytes, platelets and leukocytes based partially on this phenomenon [19].

In reproductive medicine, it was applied for the first time by Grunewald et al. in 2001 to eliminate apoptotic spermatozoa from cryopreserved semen samples [20]. Apoptotic cells show several intracellular alterations and molecular changes in the plasma membrane. One of the latter is the externalisation of phosphatidylserine (PS), which is a phospholipid located normally on the inner surface of the cell membrane [21]. Unfortunately, spermatozoa can carry this and other early signs of apoptosis without significant impact on morphology or motility, thus it can escape further phases of programmed cell death and fertilise the oocyte [22]. In a study by Hichri et al. showed that distinct apoptotic alterations (the ratio of activated caspases, externalized PS, DNA-fragmentation) are reliable predictors of ART outcomes independently of conventional sperm parameters, DNA-fragmentation with best predictive value [23]. The original method is based on the binding of PS to a protein named annexin V, showing high affinity to this molecule. Annexin V is conjugated to the surface of magnetic microbeads placed in a separation column and exposed to magnetic field. Sperm sample is loaded into the column, apoptotic spermatozoa bind to annexin V and are retained in the column (positive fraction). Non-apoptotic spermatozoa can be eluted through the column (negative fraction) and separated for further application (Figure 1) [24]. Many studies proved that MACS effectively reduces DNA-fragmentation [25,26,27], yielding spermatozoa with higher DNA integrity, meaning higher quality for ART. The technique is frequently suggested to couples with more than two ICSI failures/miscarriages with an unknown female cause with higher DNA-fragmentation by the male partner, though its application lacks any high level evidence, therefore its application should be consulted individually with each couple [22].

Since the quality of the current evidence is very low, the true clinical effects of the method on ART outcomes are still a debate and large randomized controlled studies are missing. MACS is usually not examined individually, rather combined with other techniques (mostly with SU and DGC). A Cochrane review by Lepine et al. in 2019, was inconclusive if MACS elevates clinical pregnancy and live birth rates (LBR), and reduces miscarriage rates [28]. Mei et al. in 2022 reported no significant differences in clinical pregnancy and implantation rates of the first embryo transfer cycles after MACS + SU + DGC. However, they detected a tendency to improve the LBR of the first embryo transfer cycle and the cumulative LBR along with lower number of transferred embryos and transferred number in 86 patients undergoing in vitro fertilization (IVF) and ICSI without any difference in clinical and embryological characteristics [29].

The routine application of MACS run out in the annexin V-based technique, but according to us, it hides much greater potential. Antibodies targeting different surface molecules can be attached to the microbeads and this way, we could purify the sperm/TESE sample from the contaminating cells (e.g., leukocytes, erythrocytes, epithelial cells) (Figure 1) preventing their detrimental effects (increasing ROS level, hampered visualization of spermatozoa in suspension, etc.).

## 2. Results

### 2.1. Magnetic Separation of Sperm Cells with High DNA-fragmentation

As the inclusion criteria was a high DNA-fragmentation index, the DNA-fragmentation of sperm cells before separation was 42.04 ± 9.34%. After separation, a 2.19-fold decrement was observed, from 42.04 ± 9.34% to 19.11 ± 5.90% (*p* < 1%, Figure 2).

### 2.2. Magnetic Separation of Leukocytes

In case of leukocytospermia (LCS), the initial proportion of CD45+ leukocytes was 6.92 ± 3.71% compared to the total number of cells in examined semen samples (Figure 3). After magnetic separation, the ratio of leukocytes has decreased significantly (*p* < 0.1%) to 0.44 ± 0.39%.

### 2.3. Magnetic Separation of Erythrocytes

All testicular biopsies involve some level of bleeding, which also appears in the prepared sample. The processed biopsies contained an average of 73.71 ± 39.85 M mL^−1^ erythrocytes (Figure 4). Magnetic separation of CD235a+ cells reduced noticeably the number of CD235a expressing erythrocytes from 73.71 ± 39.85 M mL^−1^ to 2.39 ± 2.04 M mL^−1^ (*p* < 0.1%, Figure 4).

## 3. Discussion

In our study, potential applications of the MACS technology were investigated. The original concept of the method is the elimination of apoptotic cells marked with annexin V, which we have successfully proven by confirming the data in the literature. After that, DNA-fragmentation is one of the inherent processes of apoptosis, so the effectiveness of magnetic separation can be measured through this. Annexin V magnetic separation of samples with elevated DNA-fragmentation resulted in a 2.19-fold decrease (*p* < 1%) in the DNA-fragmentation index (Figure 2). El Fekih et al. [30] observed a similar phenomenon, after annexin V magnetic separation, the ratio of un-fragmented, chromosomally balanced was significantly higher. These have been confirmed several times in the literature [31,32]. Esbert et al. [33] compared the apoptotic cells remaining on the separation column with the flow-through, high-quality cells and found that the rate of abnormalities was higher for each of the 17 chromosomes examined in the sample retained on the column. However, so far the results are not conclusive regarding the effect of the method on the effectiveness of IVF cycles. In some cases, they were not found any improvement on the reproductive outcomes [34].

The ejaculate basically consists of two main fractions: the spermatozoa and the seminal plasma composed of the secretions of the accessory glands (epididymides, seminal vesicles, prostate, bulbourethral glands). The semen sample ideal for ART purposes contains no leukocytes, erythrocytes or epithelial cell. On the contrary, leukocytes are present throughout the male genital tract and have important roles in the homeostasis (e.g., phagocytic clearance of abnormal or senescent spermatozoa by epididymal macrophages) [35]. In the clinical practice almost every semen sample is contaminated with leukocytes (mainly polymorphonuclear (PMN) cells (50–60%) [36]) in a different level. According to the WHO, only if their level exceeds 1.0 × 10^6^ peroxidase-positive cells mL^−1^, it has clinical significance and is considered abnormal [37]. Prevalence of leukocytospermia (LCS) or pyospermia is around 10 to 20% of infertile men [36].

In literature, a potential relationship between LCS and poorer semen quality, consequently male sub-/infertility can be found. Leukocyte infiltration can occur due to conditions with elevated cytokine levels (e.g., varicokele, smoking) [38,39] or male-accessory gland infections (MAGI; orchitis, epididymitis, prostatitis). Specific pathomechanisms and alterations in sperm parameters can be hardly correlated with particular leukocyte subpopulations, but some of them has been already observed. PMN cells activated by cytokines exhibit ROS production causing oxidative stress [40]. If the oxidative effects overwhelms the capacity of the antioxidants system of seminal plasma, lipid peroxidation, loss a membrane integrity and as a consequence, mitochondrial DNA damage and nuclear DNA-fragmentation develops in the spermatozoa [41], in addition to the deterioration of classical semen parameters (volume, concentration, progressive/total motility) and inhibition of sperm-oocyte fusion [42,43,44,45]. T-lymphocytes giving a much smaller fraction (2–5%) of seminal leukocytes can have a negative effect on sperm motility mediated by interferon-γ [46].

Haemospermia means blood appearing in the ejaculate. It can be a consequence of several pathological conditions (infections, urogenital malignancies, prostatic or vesicular cysts, etc.), though 30–70% is idiopathic [37,47]. In case of testicular sperm retrieval (TESE), the gained tissue sample contains not only the spermatozoa from the tubules, but many cells of other origin (leukocytes, erythrocytes, epithelial cells).

In both cases, the biological sample from the male partner, which we intend to use for ART, is contaminated with erythrocytes exerting potential unfavourable effects on spermatozoa. In such cases, where only a few spermatozoa are present in the sample (even immotile in case of testicular sperm), the abundant erythrocytes can make sperm isolation extremely difficult and time-consuming. To facilitate this, several methods have been already developed (erythrocyte-lysing buffer, pentoxifylline to stimulate sperm motility), but some concerns have been raised about the safety of their application and it is still a debate [48,49,50,51]. Cryopreservation of sperm for later use is a common practice in fertility centers, therefore another important aspect worth to consider is the harmful effect of erythrocytes on frozed-thawed spermatozoa through hemolysis (combined oxidative and cytotoxic effect of released heme and iron molecules) [52].

Cluster of differentiation (CD) is an identification system established in 1982 [53] based on the so-called clusters of similar cellular surface antigens recognized by antibodies which gives the possibility of identify cell subpopulations based on their immunophenotype. Currently, more than 400 proteins have been designated as CD markers [54]. CD45 also known as the leukocyte common antigen, is a membrane glycoprotein with a genetically highly conserved structure expressed on the surface of all nucleated haematopoietic cells, thus the only exceptions are mature erythrocytes and platelets. It has a key role in the regulation of immune response [55]. CD235a marks the transmembrane glycoprotein glycophorin A (GPA), its presence is restricted to erythrocytes and their precursors. It is responsible for the shape of red blood cells [56].

The modification of the original annexin V-based method using microbeads coated with anti-CD45 or anti-CD235a gives the opportunity not only to enrich the target cells (spermatozoa), but to eliminate the undesired cells (leukocytes, erythrocytes) from the sample (ejaculate, testicular tissue).

Thinking further about the basic principle of MACS separation, we investigated other possible uses, such as the separation of leukocytes and erythrocytes. Previously, the removal of contaminating leukocytes from semen samples using MACS was already examined in a few studies [57,58,59]. Ochsendorf et al. used anti-CD67 coated microbeads as well, which bind specifically to granulocytes, the most frequent subgroup of leukocyte in semen with supreme role in ROS production. The limitation of the method could be the activation of leukocytes via binding to the antibody leading to ROS release. In the study of Marietta et al. [60], CD45 magnetic beads were applied to isolate and identify different leukocyte populations in human semen successfully. Similarly to the literature, the application of CD45 magnetic beads on semen samples of patient suffering in LCSresulted in significant decrement in the ratio of CD45+ cells, in comparison with unseparated controls (Figure 4). In this way, the method can contribute to the elimination of the aforementioned negative effects in vitro. However, it has to be mentioned that the effect of leukocytes on spermatozoa in vitro may be less severe than in vivo. Adams et al. [61] described and validated a multitarget MACS method which combines microfluidics with magnetic separation on a chip-based platform providing highly selective separation of bacterial cells based on multiple surface marker. In the future, the development of a device for sperm separation based on this technology could make possible both the purification of semen sample from contaminating cells and isolation of high-quality spermatozoa in a single step process. The main advantages of MACS are its high selectivity for target cells based on immunomagnetic principle, specificity (based on good contrast between target and non-target cells), ability to control accurately the strength of the magnetic field to optimize it for the specific cell type (e.g., sperm) and integrability with other separation methods. Furthermore, it is easy-to-use, the contribution of a highly trained operator is not necessary. The main challenge using the method is the detachment of the target cells from the magnetic beads without detrimental effects on their functions and viability. It can be achieved via saturated protein solutions, enzymes, temperature/pH/electricity change/light-induced release, aptamers, hydrogels [62]. The most apparent approach to bridge this problem would be the selection of all non-target cells (negative selection), but in a clinical setting this might be also challenging considering the high variety of contaminating cells in biological samples.

There are several options available for sorting different types of cells, but basically two methods are widely used, fluorescent-activated cell sorting (FACS) and MACS. Compared to MACS, FACS sorters are more powerful and can be used to produce more pure cell populations. An additional advantage is that it allows the separation of cells according to their surface markers, as well as their size and granularity. For the purification of rare populations, multi-color staining is also possible with some instruments. It follows from all of this that FACS sorters require special skills and represent a serious financial investment for certain laboratories. Unlike FACS, available commercial MACS instruments are less sophisticated, more compact, and much less expensive. There are many benchtop magnetic cell sorters, either manual or automated. A trained technician is not required for their use, the cell sorting process is much simpler and more time-saving [63]. The above-mentioned advantages may allow MACS to spread widely in IVF laboratories, whether it is for the separation of apoptotic sperm cells, aimed at reducing DNA-fragmentation, or the selection of other components of the semen sample.

Our literature review cannot find any previous publication about the elimination of erythrocyte in semen with MACS, thus we are the first to give a description of this setting. Regarding the obstetric and perinatal outcomes, the method seems to be safe [64], but further studies are needed focusing on the potential effects of magnetic force on spermatozoa and its aftermath on ART outcomes. This is the first time, that significant (*p* < 0.1%), 30.8-fold reduction in the number of erythrocytes has been published in TESE samples with magnetic, CD235a separation.

We do not claim that annexin V, CD45 or CD235a magnetic separations will bring a breakthrough in the field of IVF or increase dramatically its success. Nevertheless, the method may be worth rethinking. It would be important to identify a molecule on spermatozoa that allows for positive/negative selections for the development of new magnetic beads.

## 4. Materials and Methods

### 4.1. Sample Collections

After 3 days of abstinence, semen samples were collected by masturbation. After liquification on 37 °C (not more than 1 h), semen analyses were performed by an SCA SCOPE (Microptic S.L., Barcelona, Spain) automatic semen analysis system and a MACSQuant Analyzer 16 flow cytometer (Miltenyi Biotec, Bergish Gladbach, Germany). In our clinic, complete sperm analysis (sperm count, motility, progressive motility, vitality, DNA-fragmentation, count of round cells and peroxidase-positive cells) is performed on all men applying for an andrological examination.

If high DNA-fragmentation index (>30%) was found during the routine examination (*n* = 17), annexin V labeling and magnetic separation was performed for research purposes. Inclusion criteria were the high DNA-fragmentation index and signed research consent form.

If LCSwas found during the routine examination (*n* = 13), magnetic separation was performed for research purposes. Inclusion criteria were confirmed LCS and signed research consent form.

In case of testicular biopsies, testicular tissues (*n* = 12) were processed mechanically. Unprocessed tissue scraps were eliminated by density gradient centrifugation. After centrifugation, sediments were used for magnetic separations. Inclusion criteria were confirmed sperm hit in the biopsy and signed research consent form.

### 4.2. Magnetic Separation of Apoptotic Sperm Cells for Decreasing DNA-Fragmentation

Separations were carried out based on the recommendations of the manufacturer. Namely, after the determination of cell count, 1–2 × 10^7^ sperm mL^−1^ was sedimented by density gradient centrifugation carried out at 300× *g* for 12 min. For density gradient centrifugation, sperm samples were processed using 40% and 80% PureCeption^TM^ (PureCeption^TM^ is a sterile colloidal suspension of silica particles stabilized with covalently bound hydrophilic silane formulated in HEPES-buffered human tubal fluid) gradient layers. Briefly, 1 mL of the lower phase gradient (80%) was moved into a conical bottom tube. A second 1 mL layer of upper phase (40%) was then slowly placed over the lower phase. PureCeption^TM^ was diluted with sperm preparation medium. A proper volume of liquefied semen was gently placed over of the upper phase. The prepared tube was then centrifuged as described above. The supernatant was discarded, cells were resuspended in MACS ART binding buffer and centrifugated at 300× *g* for 4 min. The supernatant was discarded, 200 μL of MACS ART annexin V reagent was added to cells and the final volume was completed to 500 μL with the MACS ART binding buffer. This solution was incubated for 15 min at room temperature. A MACS ART MS separator column was rinsed with MACS ART binding buffer and the labeled cells were loaded onto the column. Unlabeled cells that passed through were collected. Sperm DNA-fragmentation measurement with TUNEL assay was done before and after separations based on the method of Sharma et al. [65] DNA-fragmentation index was calculated by dividing the TUNEL-positive sperm count by the total number of sperm cells.

### 4.3. Magnetic Separation of Leukocytes

Separations were carried out based on the recommendations of the manufacturer. Namely, semen samples were centrifuged at 300× *g* for 10 min, the plasmas were discarded and the cell pellets were resuspended in 80 μL of buffer (PBS, containing 0.5% bovine serum albumin and 2 mM EDTA, pH 7.2) per 10^7^ total cells. 20 μL of CD45 (clone REA747) magnetic microbeads were added to the samples per 10^7^ total cells. After mixing, this solution was incubated for 15 min on room temperature (the manufacturer recommends 4–8 °C but for spermatozoa it is not an optimal temperature). After the incubation, 2 mL of PBS was added and a centrifugation was performed as described earlier. The supernatant was aspired completely and up to 10^8^ cells were resuspended in 500 μL of PBS. LS separator columns were rinsed with PBS and the labelled cell suspensions were loaded onto the columns. Unlabeled cells that passed through were collected. The unlabeled cell fraction was labeled with CD45-FITC stain to ensure complete removal of CD45+ cells. Flow cytometric analyses were performed. Ratios of CD45+ cells were determined before and after magnetic separations.

### 4.4. Magnetic Separation of Erythrocytes

Separations were carried out based on the recommendations of the manufacturer. Namely, prepared testicular biopsies were centrifuged at 300× *g* for 10 min, the supernatants were discarded and the cell pellets were resuspended in 80 μL of PBS buffer (the composition is the same as described above) per 10^7^ total cells. 20 μL of CD235a (clone REA175) magnetic microbeads were added to the samples per 10^7^ total cells. After mixing, this solution was incubated for 15 min on room temperature. After the incubation, 2 mL of PBS was added and centrifugation was performed as described earlier. The supernatant was aspired completely and up to 10^8^ cells were resuspended in 500 μL of PBS. LS separator columns were rinsed with PBS and the labelled cell suspensions were loaded onto the columns. Unlabeled cells that passed through were collected. Number of erythrocytes before and after magnetic separation was observed and counted with light microscopy. Erythrocytes were identified based on their colour and morphological features.

### 4.5. Chemicals

For magnetic separations, annexin V, CD45 and CD235a magnetic beads; and for fluorescent labeling, CD45-FITC fluorophore were purchased from Miltenyi Biotec. For buffer preparation, chemicals were purchased from Sigma Aldrich (Sigma Aldrich, St. Louis, MO, USA). TUNEL assay for DNA-fragmentation determination was from Thermo Fisher Scientific (Thermo Fisher Scientific, Waltham, MA, USA). For density gradient centrifugations, PureCeption^TM^ and sperm preparation medium were purchased from Origio, CooperSurgical (CooperSurgical, Trumbull, CT, USA).

### 4.6. Statistical Analysis

The data were given as the average and standard deviation. A Shapiro–Wilks test was used to evaluate the distribution of the data. Normally distributed variables were examined using the parametric one sample *t*-test. GraphPad in Stat 7.0 software (Dotmatics, GraphPad Software Inc., Boston, MA, USA) was used for statistical analysis.

## 5. Conclusions

MACS is classically used in the annexin V-based setting resulting the isolation of a higher quality sperm subpopulation by negative selection, which means the active extraction of apoptotic spermatozoa from the sample. Substituting annexin V with various molecules (mostly antibodies), we can target undesired cells of other origin as well.

In our study, we showed that with the application of CD45 magnetic beads the ratio of CD45+ leukocyte in semen samples can be significantly decreased, which could be useful when working with leukocytospermic samples. We proved that with using CD235a beads, testicular samples can be purified from contaminating erythrocytes, decreasing their concentration significantly in the sample. This method can give aid during processing TESE samples, handling haemospermic ejaculates or as a preparative measure before cryopreservation.

To find a molecule characteristic to the highest quality spermatozoa applicable in MACS could be the direction of future investigations. State of the art mass spectrometry (MS) can reveal novel biochemical factors, thus can be a valuable tool of such research. However, the MS technique is expensive and difficult to use in clinical environment. In future developments and routine diagnostic applications, a molecular biomarker or biomarker panels can be immobilized to MACS beads to create new predictive sperm viability tests. In our opinion, this routine positive discrimination procedure can be used to select the most suitable sperm for fertilisation and thus increase the efficiency of the IVF procedure.

## Figures and Tables

**Figure 1 ijms-25-03627-f001:**
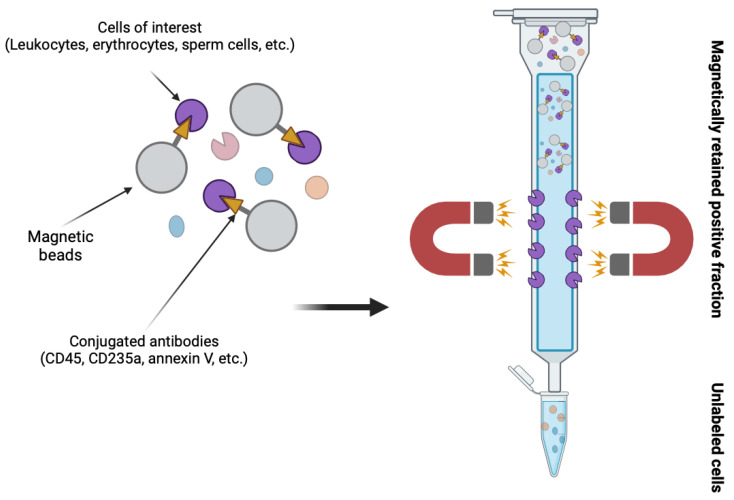
Operating principle of magnetic separation.

**Figure 2 ijms-25-03627-f002:**
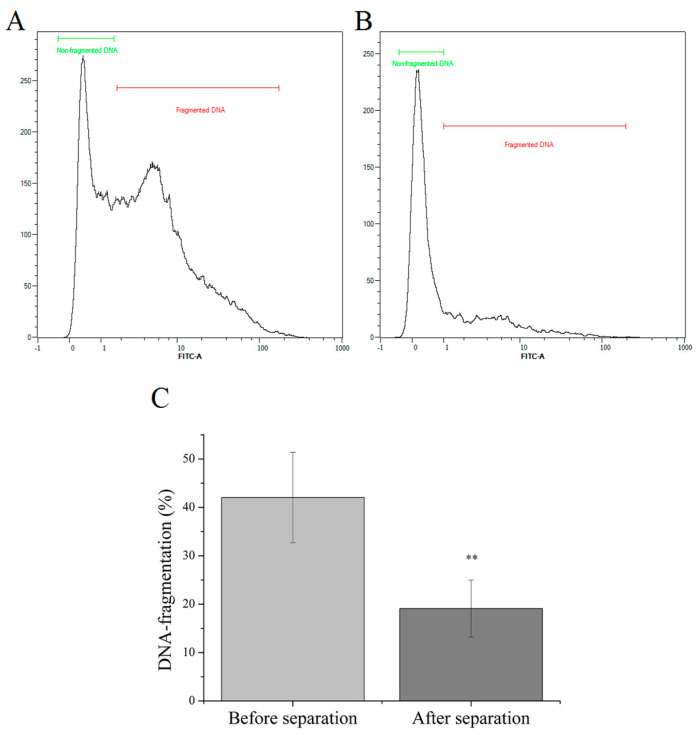
Illustrative effect of annexin V labeling and separation on sperm DNA-fragmentation. DNA-fragmentation before (**A**) and after (**B**) annexin V magnetic separation. (**C**) DNA-fragmentation indexes before and after magnetic separation of annexin V-positive sperm cells. ** *p* < 1%, *p* value was calculated via one sample *t*-test.

**Figure 3 ijms-25-03627-f003:**
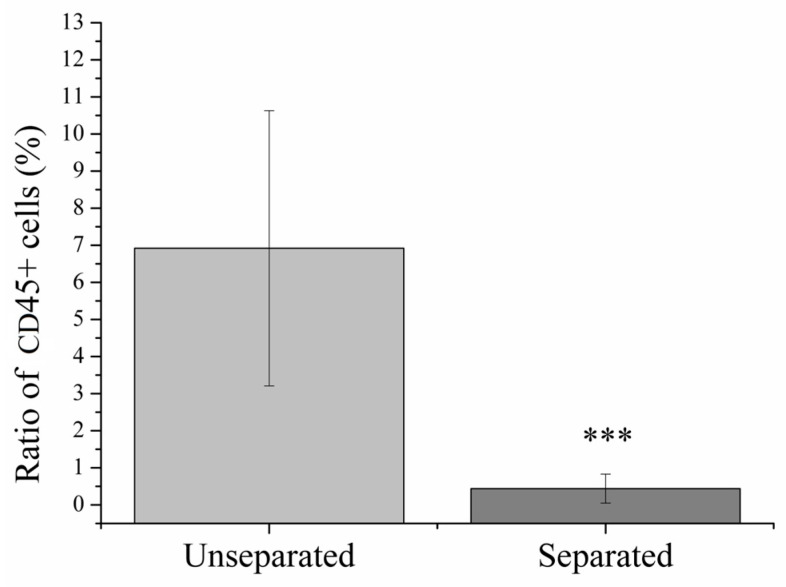
Ratio of CD45+ cells before and after magnetic separation. *** *p* < 0.1%, *p* value was calculated via one sample *t*-test.

**Figure 4 ijms-25-03627-f004:**
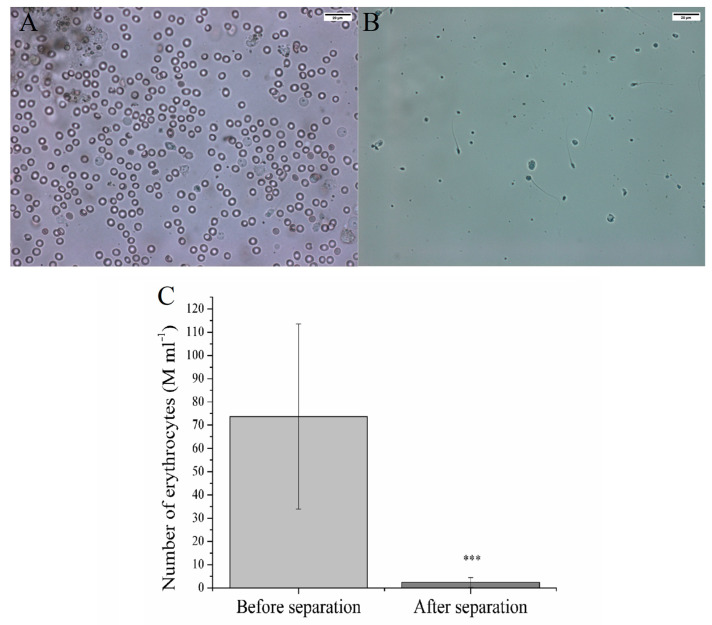
Illustrative number of erythrocytes at a testicular biopsy before (**A**) and after (**B**) magnetic separation of CD235a+ cells. (**C**) Number of erythrocytes before and after magnetic separation. *** *p* < 0.1%, *p* value was calculated via one sample *t*-test.

## Data Availability

Data are contained within the article.

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
