# Peer review of "An Alternative Application of Magnetic-Activated Cell Sorting: CD45 and CD235a Based Purification of Semen and Testicular Tissue Samples"

_ijms, 2024, doi:10.3390/ijms25073627_

Round 1

Reviewer 1 Report

Comments and Suggestions for Authors

The presented study is focused on the magnetic sorting of spermatozoa via elimination of annexin V positive cells, removal of leukocytes and erythrocytes from semen samples or samples from testicular biopsy. Authors successfully confirmed decrease in the proportion of DNA fragmented sperm cells after annexin V separation. Moreover, authors achieved significant reduction in the number of leukocytes and erythrocytes due to the elimination of CD45 and CD235a positive cells from semen and testicular samples. Anyway, it should be interesting in the future studies to observe changes in the quality of sperm cells after leukocytes removal e.g. changes in ROS level or motility parameters before and after the MACS sorting. Moreover, the effectiveness and quality of the sperm samples enhanced by such MACS sorting should be further studied in the field of IVF experiments. In conclusion, the paper is well written, and the results are clearly presented and discussed. On the other hand, some methodology issues should be clarified. For this reason, I can recommend this paper for publication in IJMS journal after a minor revision.

Below are listed minor comments that should be corrected and clarified before the final acceptation:

Results:

Line 92: How did you quantify the DNA fragmentation in sperm cells? Did you count the TUNEL positive cells with fragmented DNA as illustrated in Figure 1 or did you calculate the DNA fragmentation index (DFI)? If you calculated DFI, then please describe how.

Figure 4: Please correct “cd45” on y axe title to capitals “CD45”.

Figure 5, lines 109-111: Please explain the unit “M ml-1”.

Figure 6: If possible, please add a scalebar to microscopic images.

Discussion:

Line 162: Please correct “is labelling” to “labels” or “marks”.

Materials and Methods:

Lines 238-239: Please describe the DGC method (separation media, etc.).

Lines 242: Which solution was used for the final volume of 500 µl?

Lines 243-245: Please specify the type of used MACS sorting system.

Lines 251-252: Please correct “20 µl of CD45 magnetic microbeads”.

Line 254: Please correct “2 ml of PBS”.

Line 256: Did you label and sort 107 or 108 of spermatozoa?

Lines 256-257: Please specify the type of used MACS sorting system.

Line 259: Please specify the clone of CD45-FITC antibody.

Lines 266-267: Please correct “20 µl of CD235a magnetic microbeads”.

Line 268: Please correct “2 ml of PBS”.

Line 270: Did you label and sort 107 or 108 of spermatozoa?

Lines 270-271: Please specify the type of used MACS sorting system.

Line 273: Please specify the way of erythrocytes enumeration by light microscopy (e.g. number of analysed microscopic fields, etc.)

Line 280: Did you mean “standard” deviation?

Comments on the Quality of English Language

Some minor grammar errors.

Reviewer 2 Report

Comments and Suggestions for Authors

See attachment for my feedback

Comments on the Quality of English Language

A few minor points - see attachment
